# Modeling the Enhanced Efficacy and Curing Depth of Photo-Thermal Dual Polymerization in Metal (Fe) Polymer Composites for 3D Printing

**DOI:** 10.3390/polym14061158

**Published:** 2022-03-14

**Authors:** Jui-Teng Lin, Yi-Ze Lee, Jacques Lalevee, Chia-Hung Kao, Kuan-Han Lin, Da-Chuan Cheng

**Affiliations:** 1Medical Photon Inc., New Taipei City 242, Taiwan; 2Department of Electrical and Engineering, National Taiwan University, Taipei 100, Taiwan; r09941019@g.ntu.edu.tw; 3CNRS, Université de Haute-Alsace, F-68100 Mulhouse, France; jacques.lalevee@uha.fr; 4Department of Nuclear Medicine and PET Center, China Medical University Hospital, Taichung 400, Taiwan; d10040@mail.cmuh.org.tw; 5Department of Healthcare Administration, Asia University, Taichung City 413, Taiwan; okonkwolin@asia.edu.tw; 6Department of Biomedical Imaging and Radiological Science, China Medical University, Taichung 400, Taiwan

**Keywords:** polymerization kinetics, monomer conversion, metal composited, 3D printing, additive manufacturing

## Abstract

This article presents, for the first time, the efficacy and curing depth analysis of photo-thermal dual polymerization in metal (Fe) polymer composites for 3D printing of a three-component (A/B/M) system based on the proposed mechanism of our group, in which the co initiators A and B are Irgacure-369 and charge–transfer complexes (CTC), respectively, and the monomer M is filled by Fe. Our formulas show the depth of curing (Zc) is an increasing function of the light intensity, but a decreasing function of the Fe and photoinitiator concentrations. Zc is enhanced by the additive [B], which produces extra thermal radical for polymerization under high temperature. The heat (or temperature) increase in the system has two components: (i) due to the light absorption of Fe filler and (ii) heat released from the exothermic photopolymerization of the monomer. The heat is transported to the additive (or co-initiator) [B] to produce extra radicals and enhance the monomer conversion function (CF). The Fe filler leads to a temperature increase but also limits the light penetration, leading to lower CF and Zc, which could be overcome by the additive initiator [B] in thick polymers. Optimal Fe for maximal CF and Zc are explored theoretically. Measured data are analyzed based on our derived formulas.

## 1. Introduction

To overcome the poor mechanical properties of pure polymers, metal polymer composites (MPCs) that combine the functionalities of metals and the advantages of polymers have been proposed [1,2,3,4,5,6,7]. However, MPCs still suffer some problems such as extremely poor metal–polymer compatibility, high metal filler content, and functional singularity [8,9,10]. Light-emitting diodes (LED) photoinitiated polymerization offers many advantages such as its solvent-free formulation, mild conditions (ambient temperature, without monomer purification or stabilizers removal etc.), fast reaction rates, and alignment with the green chemistry [11,12,13,14,15]. Copper complex photoredox catalyst photocatalysts have been reported by our group for free radical/cationic hybrid photopolymerization [16,17]. Nevertheless, very few polymeric composites containing metal fillers are prepared by the photopolymerization process due to the poor light penetration depth, which also limits the curing depth.

We have recently reported rimethylolpropane triacrylate (TMPTA) as the model monomer, iron powder as the filler, and Irgacure 369 (Irg 369) as the photoinitiator with the aid of two charge–transfer complexes (CTCs) as dual thermal/photochemical initiators using LED at 405 nm, in which the lack of light penetration was indirectly overcome with the help of CTC in association with heat release during the radical photopolymerization [18,19]. In the dual thermal/photo polymerization process, we have successfully achieved 10 times more depth of curing than that of photopolymerization alone. Our experiment [19] showed that the depth of curing (Zc) is an increasing function of the light intensity but a decreasing function of the Fe and photoinitiator concentration. Furthermore, Zc is enhanced by the additive (CTC), which produces extra thermal radical for polymerization under a high temperature.

In supporting our recent measured data, Bonardi et al. [17] and Ma et al. [18,19], this article will present the kinetics, the conversion profiles features, and the depth of curing of a three-component (A/B/M) system based on the proposed mechanism of our group [19]. We will demonstrate the conversion function (CF). Analytic formulas for the CF and depth of curing and heat (temperature) profiles will be derived rigorously. Our measured data, Ma et al. [19], will be analyzed based on derived formulas and the numerical curves.

## 2. Methods and Modeling Systems

### Photochemical Kinetics

In association with the proposed scheme of our group published in Ma et al. [19], we propose, in Figure 1, a three-component system (A/B/M) defined by the co-initiators [A] and [B] with monomer (M), in which the initiator [A] is excited to its first-excited state A* and a triplet excited state T having a quantum yield (q). The triplet state T interacts with [A] to produce radical (R), which couples with M for photopolymerization and releases heat (H). Additional heat (H) is produced by the UV light absorption of the filler, iron mixed in the monomer. Heat is also produced by the light absorption of the iron filled in the monomer. The heat is transported to the additive (or co-initiator) [B] to produce extra radical R’. Therefore, the polymerization can be induced by the photo-radical (R) and enhanced by the thermal-radical (R’). A specific measured system related to Figure 1 was reported by Ma et al. [19] for a three-component system of PI/B/M, where PI is the photoinitiator Irgacure 369, B is a co-initiator of the charge–transfer complex (CTC), and the M is the monomer rimethylolpropane triacrylate (TMPTA), which is filled with filler iron (Fe).

A second example was proposed by our group published in Bonardi et al. [17], in which A is an absorbing phosphine dye of an infrared light, B is an iodonium salt as the co-initiator, and BlocBuilder MA is the thermal initiator. The light heating in the system of Ma et al. is due to the light absorption of Fe filler in the monomer TMPTA, whereas it is due to the direct light absorption of the thermal initiator in the second example. In both systems, the heat (or temperature increase) in the monomer has two components: the light heater and the heat released from the exothermic photopolymerization of the monomer.

The kinetic equations for our previous systems [20,21,22] are revised for the new system of Figure 1 (A/B/H/M) as follows. We will use the short-hand notations for the concentrations: [A] for photoinitiator Irgacure 369, [B] for the additive CTC, F for the filler iron (in wt%), and R and R’ for the photo and thermal radical, respectively.
(1)$$\frac{d\left[A\right]}{dt}=-bI\left[A\right]+REG$$
(2)$$\frac{d\left[B\right]}{dt}=-k^{\prime }\left(t\right)\left[B\right]$$
(3)$$\frac{dT}{dt}=bI\left[A\right]-\left(K^{\prime \prime }+k\left[A\right]+k^{\prime \prime }M+P^{\prime }[O_{2}\right])T\;$$
(4)$$\frac{dR}{dt}=k\left[A\right]T-KMR-P[O_{2}]R-VR^{2}$$
(5)$$\frac{{dR}^{\prime }}{dt}=k^{\prime }\left(t\right)\left[B\right]-K^{\prime }{MR}^{\prime }$$

In Equations (3) and (4), we have also included the oxygen inhibition effects (OIH), the P(z)[O_2_] term with P(z) = P_0_ exp (−Dz), and D is the oxygen diffusion constant. We note that the OIH reduces the active radical (R), leading to a delayed rising profile of the conversion efficacy for systems in air [23]. P’(z) has similar z dependence as that of P(z), i.e., near the sample surface (with small z) has higher OIH than large depth.

In Equations (2) and (5), k’(t) is the time-dependent decomposition rate of B to produce thermal radical R’, given by an Arhennius formula k’(t) = f exp[−Ed/(rH(t)]; f is the frequency factor for thermal initiator decomposition, Ed is the activation energy, and r is a gas constant. H is the heat (or temperature) in the monomer due to two components: (i) the aFI(z,t) term, due to light absorption of iron filler given by the absorption constant (a), light intensity (I), and iron concentration (F); and (ii) the heat released from the exothermic photopolymerization of the monomer (the KMR term) and (iii) the heat transport to the co-initiator, B. Therefore, H is given by a generalized heat diffusion equation
(6)$$\frac{dH\left(z,t\right)}{dt}=aFI\left(z,t\right)+KRM-b^{\prime }H+c\nabla^{2}H$$
where b’ is the heat transport constant, and the last term is the thermal conduction term in depth(z).

The UV light intensity has the depth (z) dependence due to the absorption of PI, [A], and iron filler (F) given by
(7)$$\frac{\;dI\left(z,t\right)}{dz}=-\left(b\left[A\right]+aF\right)I\left(z,t\right)$$

The monomer (M) conversion rate equation has three components: triplet state (T) direct coupling. 

Under the above quasi-steady-state solutions and for unimolecular case for R, we obtain the simplified equations (with the thermal conduction term ignored) as follows.
(8)$$\frac{d\left[A\right]}{dt}=-\left(bI\left[A\right]-REG\right)\;$$
(9)$$\frac{d\left[B\right]}{dt}=-k^{\prime }\left(t\right)\left[B\right]$$
(10)$$\frac{dH}{dt}=aFI+k\left[A\right]T-b^{\prime }H$$

The conversion efficacy (CF) defined by CF = 1 − M(t)/M_0_, with M(t) given by the solution of
(11)$$\frac{dM}{dt}=-\left[k^{\prime \prime }M+k\left[A\right]/\left(1+OIH\right)\right]\;T-k^{\prime }\left[B\right]$$

## 3. Methods and Results

A full numerical simulation is required for the solutions of Equations (9) to (12), which will be presented elsewhere. We will focus on comprehensive analysis for special features and the key factors of dual radical enhancement for efficient conversion related to the measured data of Ma et al. [19], based on the analytic formulas to be derived as follows. 

### 3.1. Analytic Results

In general, Equations (11) and (12) require numerical simulation. For analytic formulas, we further approximate g = (1 − D)/(k[A]), with D(t) = k”M(t)/(k[A]) being the second-order correction term, such that T = bI(1 − D)/(k[A]); Equations (11) and (12) are reduced to
(12)$$\frac{dH}{dt}=\left(aF+b\right)I\left(z,t\right)-b^{\prime }H-bD\left(t\right)I\left(z,t\right)$$
(13)$$\frac{dM}{dt}=-\left[1/\left(1+OIH\right)+\left(k^{\prime \prime }/k\right)M/\left[A\right]\right]\left(1-D\right)\;bI\left(z,t\right)-k^{\prime }\left(t\right)\left[B\right]$$

Analytic formulas need further assumption of RGE taken as a mean reduction factor (f’), such that bI[A] − RGE = f’bI[A] having a value of f’ = 0 to 1.0. The first-order solution of Equation (9) is given by [A] = A_0_ exp(−dt), with d = bI(z), for f’ = 1, and I(z,t) assumed to be time-independent, that is, the increase of I(z,t) due to the depletion of [A] is neglected. For the perfect recycle case, f = 0, and [A] = A_0_ is a constant. Using an average of [A(z,t)] over z and t (defined as A’), we obtain I(z) = I_0_ exp[−Gz], with G = aF + bA’.
(14)$$H\left(z,t\right)=H_{0\;\;}exp\left(-b^{\prime }t\right)+\left(d^{\prime }/b^{\prime }\right)\left[1-exp\left(-b^{\prime }t\right)\right]-Q\left(t\right)$$
where H_0_ is the initial value of H(t = 0), and d’ = (aF + b)I(z)= (aF + b)I_0_ exp[−Gz]. We have also added a correction factor Q(t) = k”M_0_/(kA_0_) [1 − exp(−D’t)]D’ = qt (for small t), with D’ = 1 + k”. We note that H(t) has a transient state value H_0_ (1 − b’t) + (d’ − q)t and a steady state value (aF + b)I(z)/b’. Moreover, for z > 0, d’ has an optimal value at F = F*, and it is a decreasing function of F, for F > F*. This optimal value is due to the competing term (aF) in d’ = (aF + b)I_0_ exp[−(aF + bA’)]. Therefore, a higher F > F* and/or A’ leads to less light penetration due to the absorption factor G = aF + bA’ in I(z) = I_0_ exp[−Gz], and also leads to lower heat released (temperature) in the monomer, H(t). 

Using Equation (15) to solve for Equation (10), with approximated k’(t) = f exp[−Ed/(rH(t)] = f[1 − Ed/(rH(t)] = f[1 − (1 − Q’t)Ed/(rH_0_)], with Q’ = (d’ − q)/H_0_ − b’, we obtain
(15)$$\left[B\right]\left(z,t\right)=B_{0\;\;}exp\left[-P\left(t\right)\right]\;$$
where P(t) = f(P’ + 0.5P”t)t, with P’ = 1 − Ed/(rH_0_), P” = Ed/(rH_0_)Q’.

Using the first-order solution of Equation (14), with D = OIH = k’(t) = 0, and M(t) = M_0_ exp(−k”dt) and [A] = A_0_ exp(−dt), with d = bI, we find the second-order solution (including k’[B]) of Equation (14) given by
M(t) = M_0_ − [dt + (k”/k)(M_0_/A_0_)V(t)] − k’ B_0_V’(t)(16)
where V(t) = [1 − exp(−st)]/s, with s = (1 + k”)bI; V’(t) = [1 − exp(−fP’t)]/(fP’), with P’ = 1 − Ed/(rH_0_).

Two special cases are analyzed for Equation (17) as follows.

(a) For steady-state: V(t) = 1/s, V’(t) = 1/(fP’), we obtain the conversion efficacy (CF) defined by CF = 1 − M(t)/M_0_,
CF(z,t) = bIt/M_0_ + k”/[(1 + k”)kA_0_] + k’ B_0_/(fP’M_0_)(17)

We note CF is proportional to the light dose (It) and that this feature is only for the case of unimolecular termination of the radical R, which is proportional to (bI). For the case of bimolecular [20], R is proportional to the power of (bI) ^0.5^, and the steady-state value of CF is proportional to (bI)^−0.5^.

(b) For transient state, V(t) = V’(t) = t, we obtain the CF
CF(z,t) = b(It)[1/M_0_ + k”/(kA_0_)] + k’tB_0_/M_0_.(18)

We note that Equation (17) has three terms proportional to the light dose E = I(z)t, and initial concentrations 1/A_0,_ 1/M_0_, and B_0_ and the absorption constants, b. However, CF (for z > 0) is a decreasing function of Fe, since I(z) = I_0_ exp[−(aF + bA’)z].

### 3.2. Depth of Curing (DoC)

As shown by Equation (7) the light intensity is an exponentially decreasing function of the depth(z) according to a Beer–Lambert law (BLL). However, for the situation of time-dependent intensity I(z,t) due to the depletion of the PI concentration, A(t) = A_0_ exp(−dt), a generalized BLL was developed by Lin [20], such that the solution of Equation (8) becomes I(z,t) = I_0_ exp[−Gz], in which G is an average of [A(z,t)] over z and t (defined as A’), given by G = aF + bA’.

A curing depth (Z_C_) is defined by when the conversion efficacy is higher than a critical value, CF > C*. Using I(z,t) = I_0_ exp[−Gz], and Equation (18) for CF, we obtain
(19)$$Z_{C}=\ln\left[\left(SI_{0}\right)/\left(C^{*}-S^{\prime }\right)\;\right]/\left(aF+bA^{\prime }\right)$$
where S = bt[1/M_0_ + k”/(kA_0_)]; and the enhancement factor, S’ = k’t(B_0_H_0_)/M_0_.

We note that Z_C_ is proportional to the light dose I_0_t, but it is a decreasing function of the effective absorption constant (aF + bA’), with A’ being the z and time-averaged PI concentration [A], given by A’. Equation (19) shows that Zc is enhanced by the additive [B] via S’ = k’t(B_0_H_0_)/M_0_, which is proportional to k’B_0_H_0_, i.e., the limited light penetration due to Fe is overcome by the additive CTC, or [B]. Equation (19) shows that Zc is proportional to ln(SI_0_) under the unimolecular case, with neglected VR^2^ term in Equation (4). For the bimolecular dominant case (with VR^2^ >> KRM), Equation (19) is revised to [21]
(20)$$Z_{C}=2\ln\left[SI_{0}{}^{0.5}/\left(C^{*}-S^{\prime }\right)\;\right]/\left(aF+bA^{\prime }\right)$$

We note that more detailed discussion and numerical results of Zc and curing time may be found in our previous work [20,22].

As shown by Equation (15), when Fe is too high, the DoC is too low to create effective polymerization, and it is limited to thin films. For lower Fe of 10% to 30%, the DoC is larger than the sample thickness, and sharp edges are produced. Moreover, high Fe leads to strong light absorption, and the light dose is smaller than the polymerization threshold at a certain sample depth.

### 3.3. General Features and New Findings

As shown by Equations (9) to (18), the following are the significant features of the [A]/[B]/M system, which was also proposed by Ma et al. [19]. Our modeling has explored the following general features, in which some new findings are not yet explored in our previous experiment of Ma et al. [23], but might be justified in our future works.

(i) As shown by Equation (6), the heat (or temperature) increase in the system has two components: (a) the aFI(z,t) term, due to light absorption of iron filler given by the absorption constant (a), light intensity (I), and iron concentration (F); and (b) the heat released (the KMR term) from the exothermic photopolymerization of the monomer.

(ii) The solution of heat H(t) given by Equation (14) has two terms, which have opposite trends in their time profiles; the first term is a decaying function, whereas the second term is an increasing profile. We note that H(t) has a transient state value H_0_ (1 − b’t) + d’t, and a steady state value (aF + b)I/b’. Therefore, for F > F*, higher F and/or A’ leads to less light penetration due to the term G = aF + bA’ in I(z) = I_0_ exp[−Gz], and lower heat released (temperature) in the monomer. This feature of optimalization is valid only for z > 0. It does not exist for surface (with z = 0). Numerical simulation will be shown later for this optimal feature.

(iii) As shown by Equations (8) and (12), (14), the monomer conversion has three components: (a) from the direct coupling of T and [A], (b) the coupling of photon-radical (R) with M, and (c) coupling of thermal-radical (R’) and M, in which R’ is an enhancement radical produced from the co-initiator, [B], under a high temperature given by the heat function, H(t). We note that the Fe filler leads to temperature increase but also limits the light penetration, leading to lower CF and DoC, which could be overcome by the additive initiator [B] for thick samples.

(iv) As shown by Equation (17), the CF has terms proportional to the light dose E = It, and initial concentrations A_0,_ H_0_ and the last term of Equation (17), k’B_0_/M_0_, which is the enhancement factor in the presence of co-initiator B. The above features are for the case with unimolecular coupling of KRM term in Equation (4). For the situation of a strong bimolecular termination, we need to include the coupling of VR^2^ in Equation (4), and leading a CF that is proportional to the (bI)^0.5^ (for transient state) and inverse proportional to (bI)^−0.5^ at steady-state, i.e., higher intensity leads to a lower steady-state CF than lower intensity. The enhancement factor is given by S’ = k’t(B_0_H_0_)/M_0_, shown in Equations (17) and (18). The CF enhancement in the presence of co-initiator B was also reported experimentally by Bonardi et al. [17].

(v) As show by Equation (19), the depth of curing (DoC), Zc, is an increasing function of the light intensity. However, there is an optimal Zc, based on the parameter of (aF + b), i.e., there is an optimal value of the iron concentration (F) that can be found mathematically by taking the derivative of Zc over d’, and let it equal 0, which can be also found numerically [20]. This optimal value is due to the competing term in d’ = (aF + b)I_0_ exp[−(aF + A’)]. The enhanced Zc is given by S’ = k’t(B_0_H_0_)/M_0_. Equation (18) also defines the minimal enhancement factor (S*) to reach a sample depth of Z’ given by the condition of when (aF + b)Z’ > ln (SI_0_/(C* − S’), or S’ > S*, with S* = C* − SI_0_ exp[−(aF + b)Z’], or k’t(B_0_H_0_)/M_0_ > C* − SI_0_ exp[−(aF + b)Z’].

(vi) Our modeling also shows that there is an optimal condition for the initiator PI concentration given by d(CF)/dF = 0, with CF given by Equation (17), which also requires numerical stimulation [20].

### 3.4. Theoretical Predictions, Numerical Data, and Future Directions

We will now present numerical data based on our derived formulas, Equation (14) for H(t) and Equation (18) for DoC as follows. We also carefully choose the ranges of each key parameters, such that their roles could be well-demonstrated.

Based on Equation (14), we show, in Figure 2, the volume (at z = 1.0 cm) temperature profiles in the monomer in the absence of PI (with b = 0), with heat produced by light absorption of Fe only (or b = 0). It shows that larger Fe mixed in the monomer leads to lower temperature profiles due to the light penetration depth decreases as shown by I(z,t) = I_0_ exp(−Gz), with G = aF + bA’, which is a strong decreasing function of G. The saturation is due to the heat transport term, b’H.

Figure 3 shows the temperature profiles (at z = 1.0 cm) in the presence of co-initiators, [A] and [B], with aFI_0_ = (0, 0.1, 0.2) (1/s), for a fixed bI_0_ = 10 (1/s), shown by curves (1, 2, 3), which have much higher temperature than the case of pure Fe absorption (with b = 0), curve-4; Curves 1 to 3 also have much faster increasing profiles due to the strong heat transport term of b’H (with b’ = 0.035), compared to the weak coupling curve-4 (with b’ = 0.002). The drop of Curves 1 to 3 is due to Q(t) = qt in Equation (14), with q = 1.5 for Figure 3.

Figure 4 shows the optimal feature for a maximal heat produced by the light absorption of Fe, based on Equation (14). The optimal feature exists only for z > 0, due to the light intensity given by I(z,t) = I_0_ exp(−Gz), with G = aF + bA’, which is a strong decreasing function of (aF + bA’), for z > 0. These features are show in Figure 4, for z = (0, 0.4, 0.5, 0.6) cm.

Based on our formula, Equation (18), we also show, in Figure 5, the DoC (Zc) as the function of various parameters. Figure 5A shows that Zc is an increasing function of S’, the enhancement factor given by S’ = k’t(B_0_H_0_)/M_0_, for a given curing threshold C* = 0.4, and fixed SI_0_ = 0.7, aF + bA’ = 1.0. Figure 5B shows that Zc is a decreasing function of the factor (aF + bA’), Figure 5C shows that Zc is an increasing function of SI_0_, which is proportional to the light dose tI_0_, and the coupling constant, b, as shown by Equation (19).

Based on our modeling predictions described above, we propose to conduct the following new measurements, which are not yet explored in [24].

(i) CF profiles for various initiator concentrations, A_0_ and B_0_, to justify our predicted enhancement factor given by S, in Equations (14) and (18).

(ii) CF profiles for various light intensity (I_0_) to justify the steady-state CF scaling law of (bI_0_)^m^, with m = 1.0 and −0.5;

(iii) curing depth (Zc) for various enhanced factor given by S’ = k’t(B_0_H_0_)/M_0_, as shown by Figure 5.

(iv) the optimal feature shown by Figure 4, for various aFI_0_, for a maximal heat produced by the light absorption of Fe.

Besides the enhancement effects discussed in the present article using an additive of CTC, there are many other strategies that have been reported by our group, including the use of multi-wavelength light and strategies of reduction the oxygen inhibition [24,25,26,27,28]. A critical review of photopolymerization for 3D bioprinting can be found in [24].

### 3.5. Analysis of Measured Data

The measured data shown in the images (figures) from Ref. [19] will be analyzed as follows, where a PI Irgacure 369 is exposed to a LED at 405 nm (0.11 W cm^−2^) for various iron (Fe) concentrations.

Image 1A from Ref. [19] shows the measured conversion profiles of TMPDA for various Fe concentrations. These profiles can be analyzed by our CF shown in Equation (18), which shows a typical saturation profile defined by Q’(t) = [1 − exp(−b’t)]/b’ in Equation (17).

Image 1B from Ref. [19] shows the measured DoC for various Fe concentrations. These profiles can be analyzed by our Zc shown in Equation (19), which shows Zc is an increasing function of the light intensity I_0_ but a decreasing function of the Fe (F) and PI concentration (A’), shown by the term (aF + bA’). The DoC is enhanced by the additive [B] via the S function, which is proportional to (bt)[(1/M_0_ − A_0_) + k’B_0_/(b’M_0_)]. Our measured data had yielded (10 times) thick composites under the same dose using a LED irradiation at 405 nm.

Image 1C from Ref. [19] shows the temperature profiles in the absence of PI. They are related to the Equations (12) and (14) with b = 0, but b’ > 0. This feature is also predicted by our theory as shown in Figure 2.

Image 1A from Ref. [19] shows the temperature profiles for various Fe concentrations. These profiles can be analyzed by Equation (14), which shows that H(t) has a transient state value H_0_ (1 − b’t) + d’t and a steady state value d’/b’, with d’ = (aF + b)I(z), with I(z) = I_0_ exp[−Gz]. Therefore, higher F and/or A’ leads to less light penetration due to the absorption factor G = aF + bA’ and also leads to a lower heat released (temperature) in the monomer, as shown by the measure curves of Image 1D of Ref. [19]. These features are also predicted by our theory, as shown in Figure 3. The CF enhancement in the presence of co-initiator B was also reported by Bonardi [17].

We found that wt10% and wt30% Fe filler resin can achieve a perfect three-dimensional structure, as shown in Image 5A and 5B from Ref. [19] with great spatial resolution for the photoproducts in the presence of the iron fillers. However, when the content of iron keeps up to 50% wt%, the pattern was not flat, and some area under irradiation was not fabricated (shown by image 5C of Ref. [19]). See more detailed of measurement method in Ref. [19]. We note that possibly metal plus polymer composites are discussed in [28].

## 4. Conclusions

We have demonstrated theoretically and justified by measured CF profiles of a three-component (A/B/M) system based on the proposed mechanism of our group, Ma et al. [19], in which the co initiators are A, Irgacure 369, and B, charge–transfer complexes (CTC), and the monomer M is TMPDA. Our formulas show the depth of curing (Zc) is an increasing function of the light intensity but a decreasing function of the Fe and photoinitiator concentration. Zc is enhanced by the additive [B], which produces an extra thermal radical for polymerization under high temperature. The heat (or temperature) increase in the system has two components: due to light absorption of iron filler, and heat released from the exothermic photopolymerization of the monomer. The heat is transported to the additive (or co-initiator) [B] to produce extra radical R’ and enhance the monomer conversion function (CF).

## Figures and Tables

**Figure 1 polymers-14-01158-f001:**
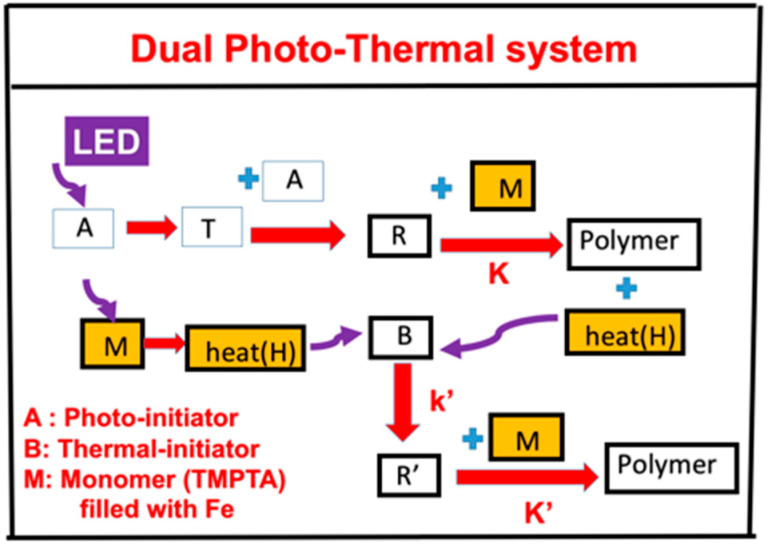
The schematics of a dual function system, (A/B/H). The heat (H) can be produced by the light absorption of the iron filled in the monomer (M) and the heat released by the photopolymerization, in which R and R’ lead to a photo-thermal dual polymerization (see text for more details).

**Figure 2 polymers-14-01158-f002:**
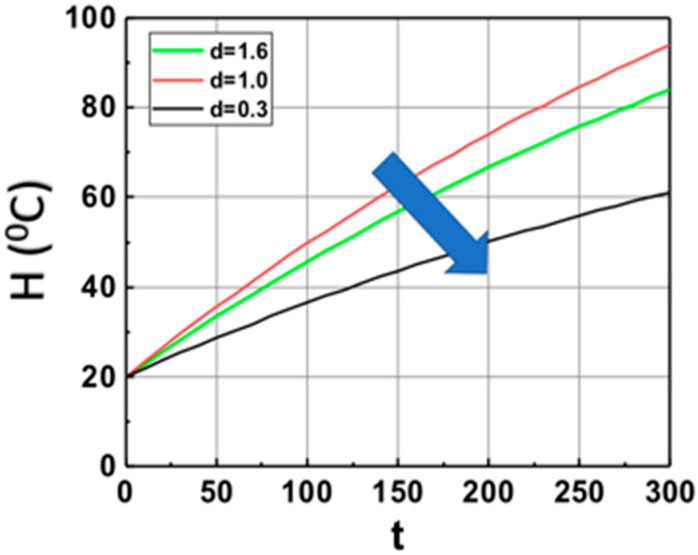
The calculated temperature (at z = 1.0 cm) profiles of monomer based on Equation (15) for various weight concentration of Fe, with aFI_0_ = (0.3, 1.0, 1.6) (1/s) and b’ = 0.002 (1/s), in the absence of PI (with b = 0), with heat produced by light absorption of Fe only, with initial temperature H_0_ = 20 °C. The arrow indicates the direction of increased values of aFI_0_.

**Figure 3 polymers-14-01158-f003:**
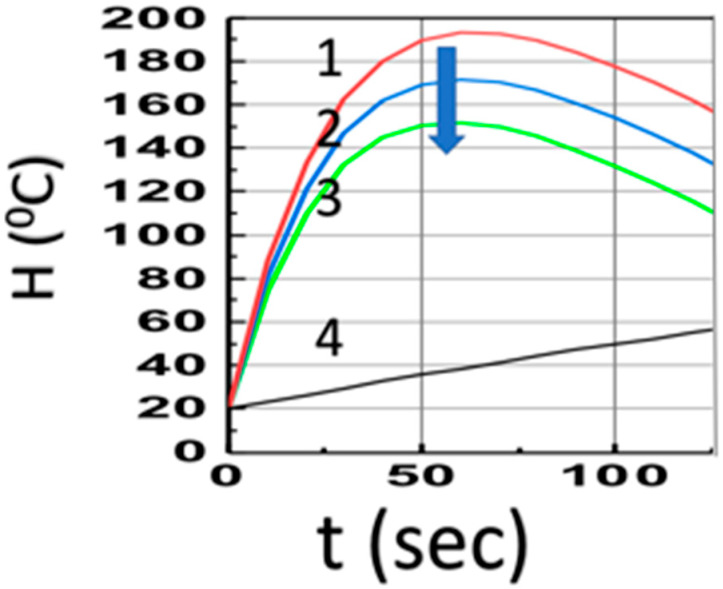
Same as Figure 2, but in the presence of co-initiators, [A] and [B], with aFI_0_ = (0, 0.5, 1.0, 1.5) (1/s), for Curves 1, 2, 3 (with bI_0_ = 10) and Curve-4 in the absence of PI (with b = 0). The arrow indicates the direction of increased values of aFI_0_.

**Figure 4 polymers-14-01158-f004:**
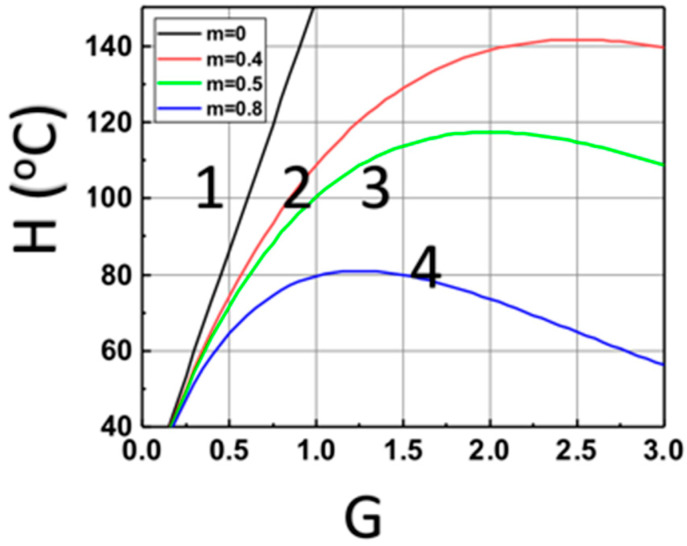
The calculated temperature of monomer versus G = aF + bA’ based on Equation (15) at various sample depth z = (0, 0.4, 0.5, 0.6) cm, for curves (1, 2, 3, 4), at t = 200 s.

**Figure 5 polymers-14-01158-f005:**
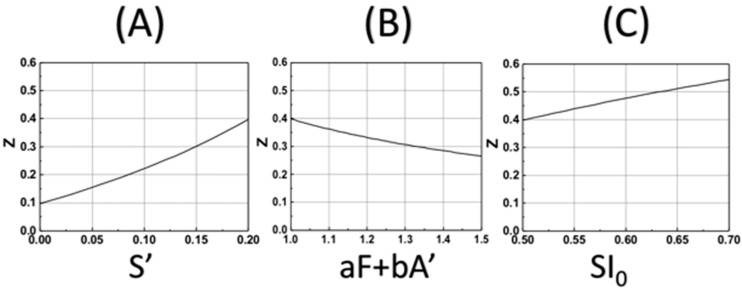
The calculated DoC (Zc) based on Equation (19) for various parameters: (**A**) for Zc vs. S’ with fixed SI_0_ = 0.7, aF + bA’ = 1.0 and C* = 0.4; (**B**) for Zc vs. aF + bA’, with fixed SI_0_ = 0.5 and S’ = 0.5; (**C**) for Zc vs. SI_0_ = 0.5, with fixed and S’ = 0.2 and aF + bA’ = 1.0.

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
