# Peer review of "Modeling the Enhanced Efficacy and Curing Depth of Photo-Thermal Dual Polymerization in Metal (Fe) Polymer Composites for 3D Printing"

_polymers, 2022, doi:10.3390/polym14061158_

Round 1

Reviewer 1 Report

In the present paper, Cheng and coworkers develop an analytical approach to describe the curing depth of iron filled polymer. The authors refer to the recently published work by the Lalevee group (see DOI: 10.1016/j.eurpolymj.2022.111011) who investigated 3 component polymerisations under 405 nm LED light combining TMPTA, Irg-369 as photo-ionisation and two charge-transfer complexes. The authors provide a kinetic model that explains the qualitative observations by Lalevee ‘s group, namely that the system’s temperature increases due to heating of the Fe filler, and exothermic polymerization, that the Fe filler hinders light penetration that which eventually can be overcome by the presence of the charge transfer complexes etc.

The derivation of the methods is relatively standard and follows the previous efforts of the authors. However, my major concern with this work is section 3.4, as the authors do not provide any clarification on who they have derived the working parameters.

Overall the manuscript is full of typos, grammatical and syntax mistakes.  This makes it difficult to read and follow. Further on in terms of structure, the authors provide section 3.5 with images completely taken from reference DOI: 10.1016/j.eurpolymj.2022.111011 which gives the first impression that the following the calculations the authors did an experiment. This is misleading and I disagree with this approach. Much of the explanation of the experimental work should be shifted to the introduction, providing motivation for this work. The chemical units and components of the previous system can be described.

The main result shown in figures 2,3, 4 should be ideally merged and the y axis correctly displayed for all in the same manner.

Author Response

Comments & Our REPLY.

(1) However, my major concern with this work is section 3.4, as the authors do not provide any clarification on who they have derived the working parameters.         REPLY: we added these text (on page 6, shown in red): We will now present numerical data based on our derived formulas, Equation (14) for H(t), and Equation (18) for DoC as follows. We also carefully choose the ranges of each key parameters, such that the roles of them could be well demonstrated.  

(2) Further on in terms of structure, the authors provide section 3.5 with images completely taken from reference DOI: 10.1016/j.eurpolymj.2022.111011 which gives the first impression that the following the calculations the authors did an experiment. This is misleading and I disagree with this approach.           REPLY: our structure was based on this;  Introduction, derive of formulas, numerical data based on formulas, then we use these formulas to analyzed our measured data, in 3.5.    In fact these measured data was done by our co-author, J. Lalavee and his group, as cited in a unpublished Ref [19]. And in order to assist the readers, we past "our" measured data and analyze them accordingly.  So, indeed we (our group) did both the calculations and the experiment, and therefore we prefer to keep the present structure.

(3) Much of the explanation of the experimental work should be shifted to the introduction, providing motivation for this work.       

 REPLY: As stated in (2), we MUST keep measurement analysis at the end section, 3.5. It is impossible to do that with referred formulas (Equations), if  section 3.5 is moved to the Introduction. However we agreed that motivations should be stated in the Introduction, so we add more text (shown in  red) in the Introduction. 

(4) The chemical units and components of the previous system can be described.      REPLY: they are complex and will take a lot of page space in this theoretical article. So, we referred the readers to read Ref [19] for more details.

(5) The main result shown in figures 2,3, 4 should be ideally merged . 

    REPLY: The parameters and roles of figs of 2,3,4 are very different. So we prefer to separate them. 

(6) the y axis correctly displayed for all in the same manner. REPLY:  yes, fixed. 

Reviewer 2 Report

This is a worthwhile manuscript – while several modifications are recommended:

The Abstract begins with “The synergic This article presents ...”

What are “metal polymer composites” ?  Possibly metal + polymer composites are meant. Composites of all kinds are discussed in W. Brostow & H.E. Hagg Lobland, Materials: Introduction and Applications, John Wiley & Sons – a reference worth consulting and citing.

“very few polymeric composite” is not in English. “two terms which has opposite trends” is not in English either.

We read “analytic formulas, Equation (14)”;  there is only one formula in that equation.

Author Response

Comments and our REPLY:

 (1) The Abstract begins with “The synergic This article presents ...” 

         REPLY:  fixed. The synergic was removed.

(2) What are “metal polymer composites” ?  Possibly metal + polymer composites are meant. Composites of all kinds are discussed in W. Brostow & H.E. Hagg Lobland, Materials: Introduction and Applications, John Wiley & Sons – a reference worth consulting and citing.  

        REPLY:  text was added on page 9 to include this Ref. [28].

(3) “very few polymeric composite” is not in English. “two terms which has opposite trends” is not in English either.

REPLY:  fixed as: composites  ,,,;        which have.

(4) We read “analytic formulas, Equation (14)”;  there is only one formula in that equation.

REPLY:  fixed as: formula,            

Round 2

Reviewer 1 Report

The authors have answered my concerns positively. Overall the manuscript is in good standing. It also reads better. I have noticed that the authors have made significant improvements in the readability. There is still minor grammar/syntax in corrections, like the ones selected below. My suggestion is to proofread at least one more time, but overall it is good to go. 

"MPCs are still suffer some"  - the verb "are" should be removed

"Light emitting diodes (LED) photoinitiated polymerization offer" - "offer" should be offers 

"(iv) As shown by Equation (17) has terms proportional " "what" has term? 

Author Response

comments fixed accordingly, and 

As shown by Equation (17), the CF has terms ....